# Involvement of Angiopoietin 2 and vascular endothelial growth factor in uveitis

**Kayo Suzuki**[1], **Daiju Iwata**[1]*, **Kenichi Namba**[1], **Keitaro Hase**[1], **Miki Hiraoka**[2], **Miyuki Murata**[1], **Nobuyoshi Kitaichi**[2], **Richard Foxton**[3], **Susumu Ishida**[1]

1 Faculty of Medicine and Graduate School of Medicine, Department of Ophthalmology, Hokkaido University, Sapporo, Hokkaido, Japan, 2 Department of Ophthalmology, Health Sciences University of Hokkaido, Sapporo, Hokkaido, Japan, 3 Roche Pharma Research and Early Development, Roche Innovation Center, Basel, Switzerland

* d.iwata@med.hokudai.ac.jp

## Abstract

### Purpose

Angiopoietin (Ang) 2 is released from vascular endothelial cells by the stimulation of vascular endothelial growth factor (VEGF)A. Ang2 increases the expression of leukocyte adhesion molecules on endothelial cells via nuclear factor κB. The aim of this study was to evaluate the effects of Ang2 and VEGFA on ocular autoimmune inflammation.

### Methods

We measured the concentrations of Ang2 and VEGFA in vitreous samples among patients with uveitis. Vitreous samples were collected from 16 patients with idiopathic uveitis (uveitis group) and 16 patients with non-inflammatory eye disease (control group). Experimental autoimmune uveoretinitis (EAU) was induced in B10.BR mice with a human interphotoreceptor retinoid-binding protein-derived peptide. The retinochoroidal tissues of the EAU mice were removed, and the mRNA levels of Ang2 and VEGFA were examined. EAU mice treated with anti-Ang2, anti-VEGFA, a combination of anti-Ang2 and anti-VEGFA, anti-Ang2/VEGFA bispecific, or IgG control antibodies were clinically and histopathologically evaluated.

### Results

The protein levels of Ang2 and VEGFA were significantly higher in the vitreous samples of patients with uveitis than in controls (P<0.05). The retinochoroidal mRNA levels of Ang2 and VEGFA were significantly upregulated in EAU mice compared to controls (n = 6, P<0.05). Although there was no significant difference, treatment with anti-VEGFA antibody reduced the clinical and histopathological scores. However, treatment with anti-Ang2 antibody reduced the clinical and histopathological scores (n = 18–20, P<0.05). Furthermore, these scores were further decreased when treated by inhibiting both Ang2 and VEGFA.

**Data Availability Statement:** All relevant data are within the manuscript and its Supporting information files.

**Funding:** This work was supported by Scientific Research by Japanese Society for the Promotion of Science [grant number 16K29294] and a grants of research by AMO Japan (Johnson & Johnson surgical vision) [grant number AS2021A000044497]. The funders had no role in study design, data collection and analysis, decision to publish, or preparation of the manuscript. F. Hoffmann-La Roche Ltd. provided the mouse cross-reactive tool antibodies used in the experimental autoimmune uveoretinitis experiments described here. F. Hoffmann-La Roche Ltd provided support in the form of salaries for their organisation's author [RF], and he was involved in elements of the study design, and preparation of the manuscript. The specific role of this author is articulated in the 'author contributions' section."

**Competing interests:** RF is an employee of F. Hoffmann-La Roche Ltd. F.Hoffmann-La Roche provided support in the form of salary for the organisation's author [RF], and he had a role in elements of the study design, and preparation of the manuscript, but no additional roles in the data collection and analysis and decision to publish the manuscript. The specific roles of authors are articulated in the 'author contributions' section. DI awarded Grants-in-Aid for Scientific Research by Japanese Society for the Promotion of Science [grant number 16K29294] and a grant of research by AMO Japan (Johnson & Johnson surgical vision) [grant number AS2021A000044497]. This does not alter our adherence to PLOS ONE policies on sharing data and materials.

## Conclusions

Based on these results, VEGFA and Ang2 were shown to be upregulated locally in the eye of both uveitis patients and models of uveitis. Dual inhibition of Ang2 and VEGFA is suggested to be a new therapeutic strategy for uveitis.

## Introduction

Uveitis is a sight-threatening ocular disease with various origins. The most common cause of uveitis in Japan is immune-related disorders, including sarcoidosis, Vogt-Koyanagi-Harada disease, and Behçet's disease [1,2]. Topical or systemic administration of corticosteroids is the mainstay of treatment for uveitis, and immunosuppressants or biologics are also prescribed. However, some cases result in irreversible visual dysfunction, even if these treatments are applied. Therefore, the development of novel therapeutic methods for uveitis is still required.

Experimental autoimmune uveoretinitis (EAU) is an animal model of endogenous human uveitis. It can be induced by immunization of mice or rats with retinal antigens, such as inter-photoreceptor retinoid-binding protein (IRBP) and S-antigen (arrestin-1), or by adoptive transfer of retinal Ag-specific T lymphocytes [3–5]. The EAU model has been widely used to investigate the pathogenesis of ocular autoimmune inflammation and to evaluate the therapeutic effects of various agents for uveitis.

Angiopoietins (Angs) are a family of secreted factors comprising Ang1 and Ang2 [6,7]. Ang1 is an agonist for tyrosine kinase with the immunoglobulin and epidermal growth factor homology domain (Tie) 2. Ang1-Tie2 signaling downregulates inflammation by blocking nuclear factor κB (NF-κB). By contrast, Ang2 has been described as a competitive antagonist interfering with Ang1-Tie2 signaling [8].

Ang2 and vascular endothelial growth factor (VEGF) A are well-known molecules that play critical and coordinated roles in pathological angiogenesis and vascular leakage [9]. Treatments targeting Ang2 and VEGFA have been developed for patients with ocular neovascular diseases [10]. Recently, these molecules have been reported to be implicated in some autoimmune inflammatory diseases, such as psoriasis, arthritis, and multiple sclerosis [11–15]. However, the roles and contribution of these molecules in ocular inflammatory diseases remain unknown. Therefore, the aim of this study is to evaluate the involvement of Ang2 and VEGFA in the pathogenesis of ocular inflammatory diseases.

## Material and methods

### Vitreous and serum samples

Vitreous and serum samples obtained from patients diagnosed with uveitis who visited the Intraocular Inflammation Clinic of Hokkaido University Hospital between 2014 and 2019 were examined in this study. Diagnosis was established by meticulous observations of ocular involvement, clinical course, and concomitant systemic diseases combined with laboratory analyses. Patients with uveitis whose etiologic diagnosis could not be established even after diagnostic vitrectomy were categorized as idiopathic uveitis.

The vitreous samples were evaluated from 16 patients with idiopathic uveitis and 16 patients with non-inflammatory ocular diseases (i.e., macular hole and epiretinal membrane) who underwent vitrectomy as controls. Their clinical characteristics are summarized in Table 1. All patients with idiopathic uveitis had marked vitreous haze, and underwent

**Table 1. Characteristics of the patients for vitreous sample.**

|  | Uveitis N = 16 | Control N = 16 |  |
|---|---|---|---|
| Male/Female | 6/10 | 8 / 8 | cases |
| Average age | 71.9 ± 12.1 | 64.5 ± 9.6 | years |

medically required vitrectomy because they were suspected masquerade syndrome such as vitreoretinal lymphoma.

We also examined serum samples from 32 patients with etiology-proven uveitis (8 with sarcoidosis, 8 with Vogt-Koyanagi-Harada disease, 8 with Behçet's disease, 8 with HLA-B27-positive acute anterior uveitis) and 8 with non-inflammatory ocular diseases as controls. Their clinical characteristics are summarized in Table 2. All patients were in disease active phase and were not receiving any treatments including corticosteroids.

This study was conducted in accordance with the guidelines of the Declaration of Helsinki and approved by the institutional review board of Hokkaido University Hospital (No. 019–0173). Informed consent was obtained from all patients enrolled in this study by the opt-out method on a website explaining the procedures performed and the reviews of medical records from Jun 2020 to Dec 2020. The medical records were reviewed and the data were collected from Jun 2021 to Mar 2022.

## Cytokine assay for vitreous and serum samples

Human vitreous samples (500–1000 μl) were collected during 25G pars plana vitrectomy prior to intraocular fluid infusion, as previously described [16], and the samples were stored at -80˚C. The samples were defrosted and centrifuged at 800 × g for 10 minutes. The supernatants were used for the quantitative immunoassay.

Undiluted serum samples were collected during their first visits. The samples were stored at -80˚C. The samples were defrosted and centrifuged at 800 × g for 10 minutes. The supernatants were used for the quantitative immunoassay.

The Ang1, Ang2, VEGFA, tumor necrosis factor (TNF) -α, interferon (IFN) -γ, and interleukin (IL) -17 protein levels of the vitreous and serum samples were measured using a magnetic multiplex bead-based quantitative immunoassay (Magnetic Luminex Assay; R&D Systems, Minneapolis, MN, USA). When the concentrations of the raw data were below the smallest value of the standard curve (Ang1: 24.84pg/ml, Ang2: 31.45pg/ml, VEGFA: 2.25pg/ml, TNF-α: 3.07pg/ml, IFN-γ: 14.58pg/ml, and IL-17: 14.10pg/ml), they were coded as 0 and were included in the statistical analysis.

## Experimental animals

Six- to eight-week-old female B10.BR (H-2$^k$) mice were obtained from Japan SLC (Japan). All animals received food and water ad libitum, in a 12-hour day/night cycle, temperature-

**Table 2. Characteristics of the patients for serum sample.**

|  | Uveitis N = 32 | | | | Control N = 8 |  |
|---|---|---|---|---|---|---|
|  | **Sar** | **VKH** | **BD** | **AAU** |  |  |
| Male / Female | 3/ 5 | 4/ 4 | 5/ 3 | 6/ 2 | 3 /5 | cases |
| Average age | 56.3±8.4 | 56.6±8.2 | 47.5±3.6 | 50.3±9.7 | 56.4±8.4 | years |

Sar; sarcoidosis, VKH; Vogt-Koyanagi-Harada disease, BD; Behçet's disease, AAU; acute anterior uveitis.

controlled environment. All studies were conducted in compliance with the Association for Research in Vision and Ophthalmology Statement for Use of Animals in Ophthalmic and Vision Research and approved by the Ethic Review Committee for Animal Experimentation of Hokkaido University (No. 16–0082).

## Reagents

The K2 peptide (ADKDVVVLTSSRTGGV; molecular weight, 1603.78), corresponding to the amino acid sequence 201–216 of bovine IRBP, is the immunodominant retinal autoantigen of EAU in H-$2^k$ mice [17], which was synthesized by Sigma-Aldrich Japan (Tokyo, Japan). Purified *Bordetella pertussis* toxin (PTX) was obtained from Sigma-Aldrich (St. Louis, MO, USA). Complete Freund's adjuvant (CFA) and *Mycobacterium tuberculosis* strain H37Ra were purchased from Becton Dickinson (Franklin Lakes, NJ, USA). Anti-Ang2/VEGFA bispecific antibody, anti-Ang2 antibody, anti-VEGFA antibody, and non-specific monovalent antibody were kindly provided by F.Hoffmann-La Roche Ltd. (Basel, Switzerland).

## Immunization of EAU

EAU was induced in the B10.BR (H-$2^k$) mice by subcutaneous injection of 100 nmol K2 peptide emulsified in CFA containing 2.5 mg/ml *M tuberculosis* strain H37Ra. Injections were made in the upper back and flanks, followed concurrently by the intraperitoneal injection of 0.1 µg of PTX in 100 µl of PBS. To evaluate mRNA expression ratios, control mice were immunized by subcutaneous injection of the emulsion of CFA containing 2.5 mg/ml *M. tuberculosis* strain H37Ra and PBS without K2 peptide.

## Treatment

Monovalent anti-VEGFA antibody, monovalent anti-Ang2 antibody, a combination of monovalent anti-Ang2 and anti-VEGFA antibodies, or anti-Ang2/VEGFA bispecific antibody was administered intraperitoneally at a dose of 20 mg/kg body weight five times in total, every 5 days throughout the experiment, from the day before the induction of EAU. A monovalent isotype control IgG antibody was administered at the same frequency as the control group. All mouse cross-reactive tool antibodies used here have been generated as CrossMabs [18] with an IgG-like shape using different combinations of binders. In the bispecific antibody, an anti-VEGFA antibody was combined with an anti-Ang2 antibody within the CrossMab format, whereas for monovalent controls either the anti-VEGFA or the anti-Ang2 was replaced with an antibody that does not bind to protein.

## mRNA levels in the retina and choroid of EAU

The EAU and control mice were euthanized with an overdose of anesthesia by intraperitoneal injection of Pentobarbital (0.5ml, 1.2%), and their retinochoroidal tissues were extracted on Days 11, 16, and 21 after immunization.

The expression levels of Ang1, Ang2, and VEGFA mRNA in the retinochoroidal tissues were examined using quantitative real-time PCR. Total RNA was extracted from cells using NucleoSpin RNA plus (Macherey-Nagel, Dueren, Germany) and reverse transcribed to cDNA using GoScript reverse transcriptase (Promega, Fitchburg, WI, USA), according to the manufacturers' protocol. Analysis of mRNA levels was performed on a StepOnePlus Real-Time PCR System (Thermo Fisher Scientific, Waltham, MA, USA) using GoTaq qPCR Master Mix (Promega, USA).

The primer sequences used for real-time PCR and the expected size of the amplification products were as follows: 5′-ACCATTTCGAGACTGTGCAGAT-3′ (forward) and 5′-CTGTCCA ACCTCCCCCATTC-3′ (reverse) for Ang1; 134 bp, 5′-CATCAGCCAACCAGGAAGTG-3′ (forward) and 5′-AAGGACCACATGCGTCAAAC-3′ (reverse) for Ang2; 112 bp, 5′-CCACGACAGA AGGAGAGCAGA-3′ (forward) and 5′- GCAGTAGCTTCGCTGGTAGAC-3′ (reverse) for VEGFA; 71 bp.

## Clinical and histopathological examination of EAU

The clinical score of EAU was evaluated by funduscopy every 3 or 4 days starting from 7 days after immunization under anesthesia by intraperitoneal injection of Pentobarbital (0.1ml, 1.2%). Clinical severity was graded on a five-point scale based on the extent of vessel dilatation, vessel white focal or linear lesions, retinal hemorrhage, and retinal detachment, as described previously [19,20]. Clinical evaluations were performed by two ophthalmologists (K.S., and D.I.) in a masked fashion.

On Day 21 after the induction of EAU, the mice were euthanized with an overdose of anesthesia by intraperitoneal injection of Pentobarbital (0.5ml, 1.2%). The eyes were enucleated and fixed with SUPER FIX (Kurabo, Osaka, Japan) for the preparation of paraffin sections. Fixed tissues were stained with hematoxylin and eosin. The histopathological severity score was graded on a scale of 0–4, as described previously [21].

The histopathological severities of EAU mice were graded by one of the authors (N.K.) in a masked fashion. During the study, the only drug-administering researcher (K.S.) had the identity of treatment animals received.

## Statistical analysis

The concentrations of Ang1, Ang2, VEGFA, TNF-α, IFN-γ, and IL-17 in the vitreous samples were statistically analyzed using Mann-Whitney U test. The concentrations of Ang1, Ang2, VEGF, TNF-α, IFN-γ, and IL-17 in the serum samples of patients with Sar, VKH, BD, AAU, and controls were statistically analyzed using Kruskal–Wallis tests and Dunn procedure, and with uveitis and controls were statistically analyzed using Mann-Whitney U test. The mRNA levels in the retinochoroidal tissues of the EAU mice were calculated by the ΔΔCt method with the level of Hprt1 mRNA as a normalization control, and statistically analyzed using the Mann–Whitney U test. Clinical and histopathological scores were statistically analyzed using Kruskal-Wallis tests and Dunn procedure.

All data were analyzed using StatPlus software version v7 (AnalystSoft Inc.) and IBM SPSS Statistics (IBM Corp.). A p-value of less than 0.05 was considered significant.

## Results

### Protein levels of cytokines in the vitreous samples of the patients with uveitis

To evaluate the pathogenesis of uveitis, we measured the protein levels of Ang1, Ang2, VEGFA, TNF-α, IFN-γ, and IL-17 in vitreous humor samples of uveitis patients.

The protein levels of Ang1, Ang2, VEGFA, TNF-α, IFN-γ, and IL-17 in the uveitis group (117.2 ± 26.9 ng/ml, 194.4 ± 122.6 ng/ml, 38.0 ± 73.5 pg/ml, 13.0 ± 2.1 pg/ml, 80.8 ± 30.4 pg/ml, and 15.7 ± 2.5 pg/ml, respectively) were significantly higher than in the control group (78.4 ± 9.7 ng/ml, 60.6 ± 11.79 ng/ml, 17.0 ± 6.0 pg/ml, 6.9 ± 0.3 pg/ml, 23.4 ± 1.1 pg/ml, and 10.4 ± 0.2 pg/ml, respectively) (Fig 1; P<0.05). In addition, the protein level of Ang2 was significantly higher than that of Ang1 in the vitreous samples of the uveitis group (P<0.05),

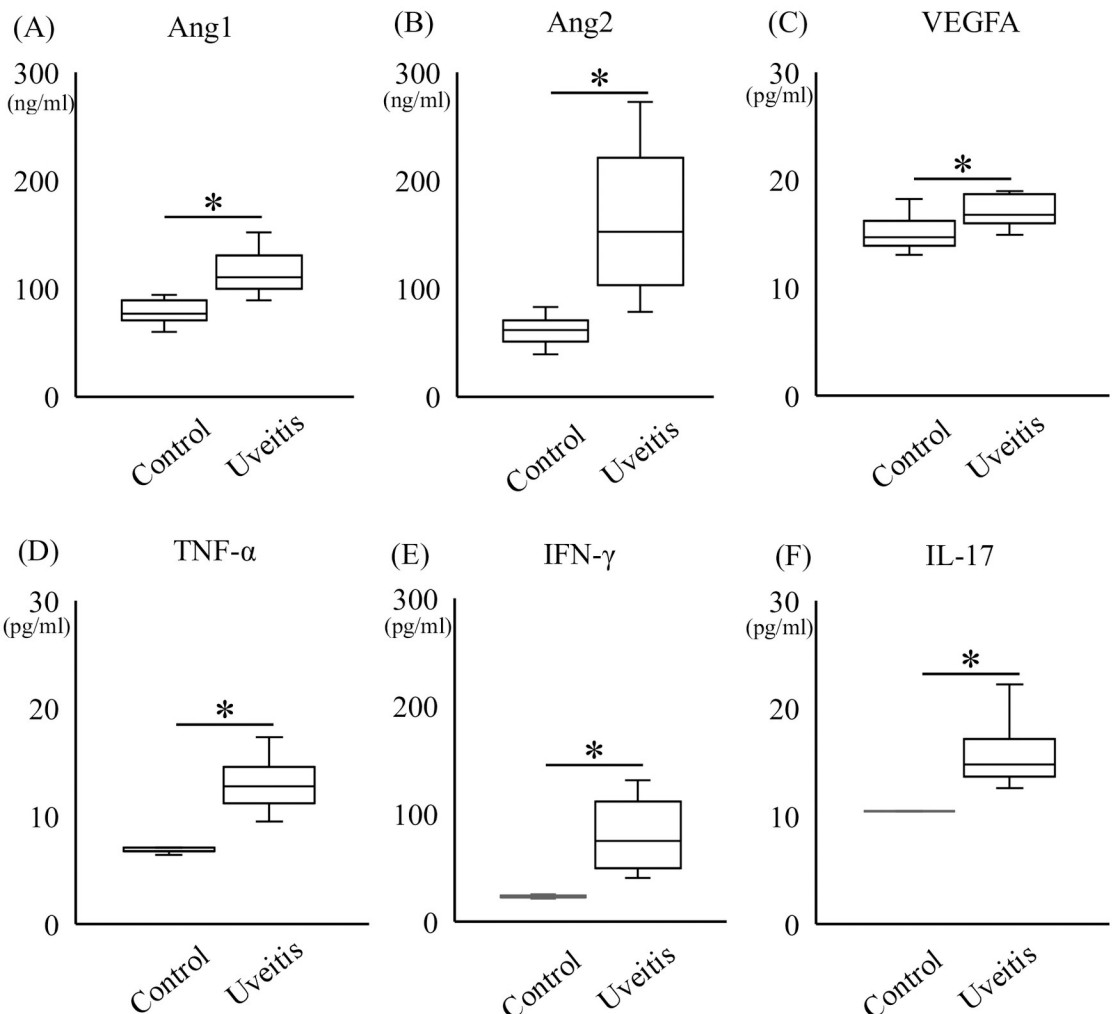

**Fig 1. Vitreous levels of Ang1, Ang2, VEGFA, TNF-α, IFN-γ, and IL-17.** The protein levels of Ang1 (A), Ang2 (B), VEGFA (C), TNF-α (D), IFN-γ (E), and IL-17 (F) in the vitreous samples of uveitis group were significantly higher than in controls (n = 16 for uveitis and n = 16 for control) by Mann-Whitney U test, *P<0.05.

whereas Ang2 was significantly lower than Ang1 in the control (P<0.01). Each ratio of Ang2/Ang1 for uveitis group and control group were 1.69 ± 1.14 and 0.77 ± 0.10, respectively.

## Protein levels of cytokines in the serum samples of the patients with uveitis

The protein levels of five groups (8 with sarcoidosis, 8 with Vogt-Koyanagi-Harada disease, 8 with Behçet's disease, 8 with HLA-B27-positive acute anterior uveitis and 8 with non-inflammatory ocular diseases as controls) each were compared, but no significant differences were found (P = 0.45–0.96).

Since there were no significant differences between the different uveitis groups, uveitis patients were pooled together (n = 32) for statistical analysis and compared with controls (n = 8). The protein levels of Ang1, IFN-γ, and IL-17 in the uveitis group (6.8 ± 5.8 ng/ml, 89.9 ± 21.3 pg/ml, and 12.0 ± 1.8 pg/ml, respectively) were significantly higher than those in the control group (2.5 ± 1.3 ng/ml, 75.2 ± 14.8 pg/ml, and 10.4 ± 0.9 pg/ml, respectively)

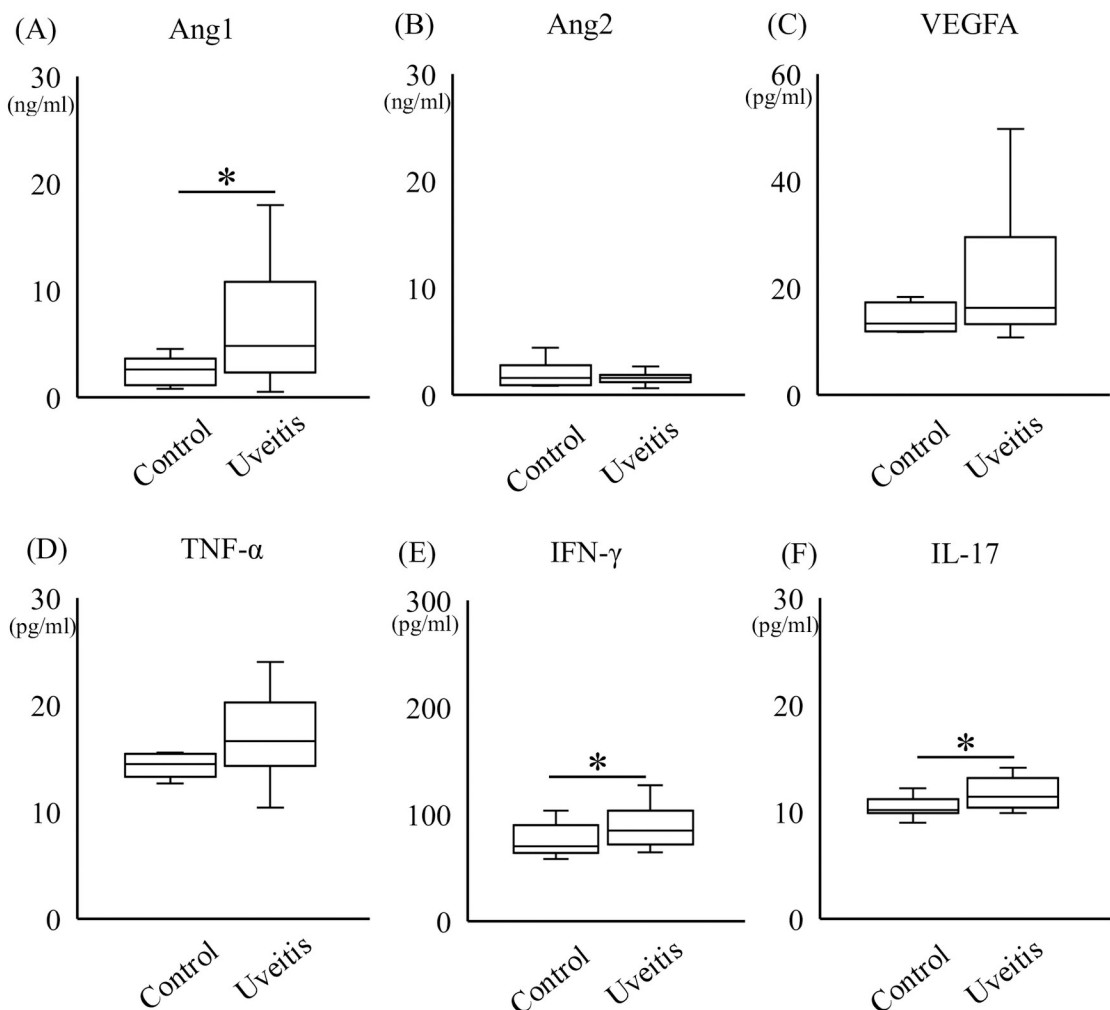

**Fig 2. Serum levels of Ang1, Ang2, VEGFA, TNF-α, IFN-γ, and IL-17.** The protein levels of Ang1 (A), Ang2 (B), VEGFA (C), TNF-α (D), IFN-γ (E), and IL-17 (F) in the serum samples of uveitis group were evaluated in the patients with uveitis and controls (n = 32 for uveitis and n = 8 for control). The protein levels of Ang1, IFN-γ, and IL-17 in the uveitis group were significantly higher than those in the control group. The levels of Ang2, VEGFA, and TNF-α had no significant differences between the uveitis group and the control group. The significance was determined by Mann-Whitney U test, *$P<0.05$.

(Fig 2; $P<0.05$). The levels of Ang2, VEGFA, and TNF-α were not significantly different between the uveitis group and the control group (P = 0.16–0.91). In addition, Ang1 was significantly higher than Ang2 in the uveitis group ($P<0.05$), whereas no difference was found between Ang1 and Ang2 in the controls.

## mRNA levels of Ang2 and VEGFA were upregulated in EAU mice

Changes in the mRNA levels of Ang1, Ang2, and VEGFA of retinochoroidal tissues derived from EAU mice or control mice immunized without K2 peptide were examined during the time course of EAU.

As shown in Fig 3 (n = 4–6 per groups), no significant difference was found in the mRNA level of Ang1 between the EAU mice and the control mice throughout the time course (P = 0.27–1.00). In contrast, the mRNA level of Ang2 was significantly higher in the EAU mice

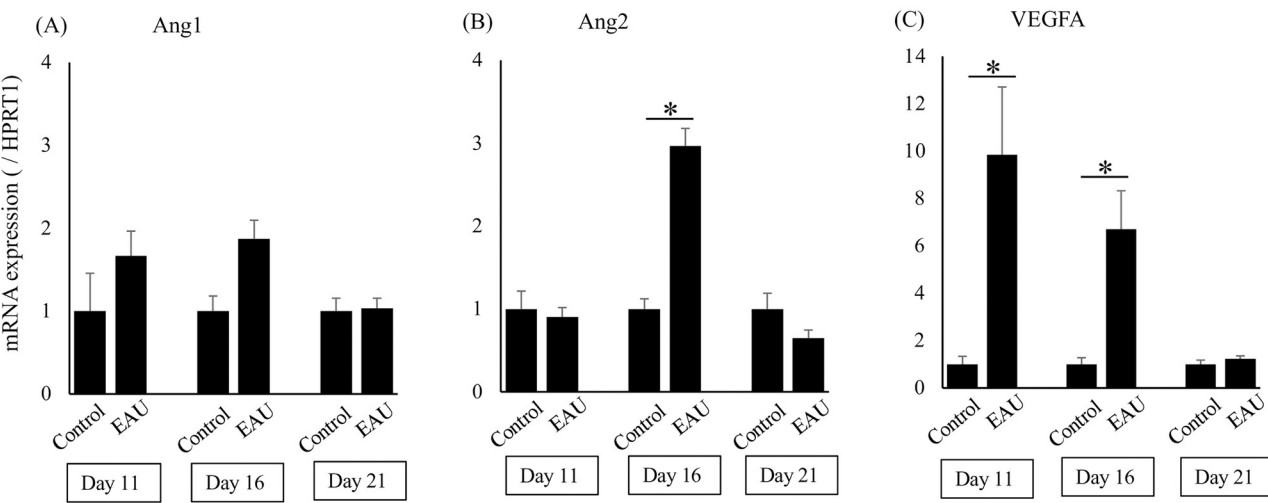

**Fig 3. mRNA levels of Ang1, Ang2, and VEGFA.** No significant difference was found in the mRNA level of Ang1 (A) between the EAU mice and the control mice throughout the time course (n = 4–6 per groups). The mRNA level of Ang2 (B) was significantly higher in the EAU mice than in the controls on Day 16, P<0.05. The mRNA level of VEGFA (C) was significantly higher in the EAU mice on Days 11 and 16, P<0.05. Results are presented as Mean ± SEM, and significance was determined by Mann-Whitney U test.

than in the controls on Day 16 (P<0.05). The mRNA level of VEGFA was significantly higher in the EAU mice on Days 11 and 16 (P<0.05).

## Amelioration of EAU by simultaneous blockade of Ang2 and VEGFA in mice

The clinical scores during the time course of EAU are shown in Fig 4 (n = 18–20 per treatment group). The experiment was performed with n = 20 mice per group, however; the control IgG antibody group and anti-VEGF antibody group were analyzed with n = 18 because some mice were no longer evaluable during the course of the experiment. Although there was not significant difference, the score of those treated with anti-VEGFA antibody was lower than that of those treated with control IgG antibody (P = 0.07–0.52). The score of the anti-Ang2 antibody treatment group was significantly lower on Days 14 (P<0.01), 17 (P<0.05), and 21 (P<0.05). Furthermore, the dual inhibition groups (i.e., the group treated with the combination of anti-Ang2 antibody and anti-VEGFA antibody and the group treated with Anti-Ang2/VEGFA bispecific antibody) showed a similar clinical course during the observation period, and these scores were significantly lower than those from mice treated with control IgG antibody on Days 14 (P<0.05), 17 (P<0.01), and 21 (P<0.01).

The histopathological scores on Day 21 after immunization of EAU are shown in Fig 5A (n = 18–20). No significant difference was found between the histopathological score of the anti-VEGFA antibody treated group (0.9 ± 0.8) and that of the control group (P = 0.12), although the score of anti-VEGFA antibody treatment group showed a trend toward being reduced. The histopathological score of EAU mice in the anti-Ang2 antibody treatment group (0.8 ± 0.6) was significantly lower than that of those treated with control IgG antibody (P<0.05). The histopathological scores of the EAU mice in the dual inhibition groups (i.e., anti-Ang2 antibody with anti-VEGFA antibody treatment group [0.5 ± 0.4] and anti-Ang2/VEGFA bispecific antibody treatment group [0.4 ± 0.6]) were significantly lower than that of those in controls (1.4 ± 0.7) (P<0.01). In the histopathology of EAU mice treated with anti-

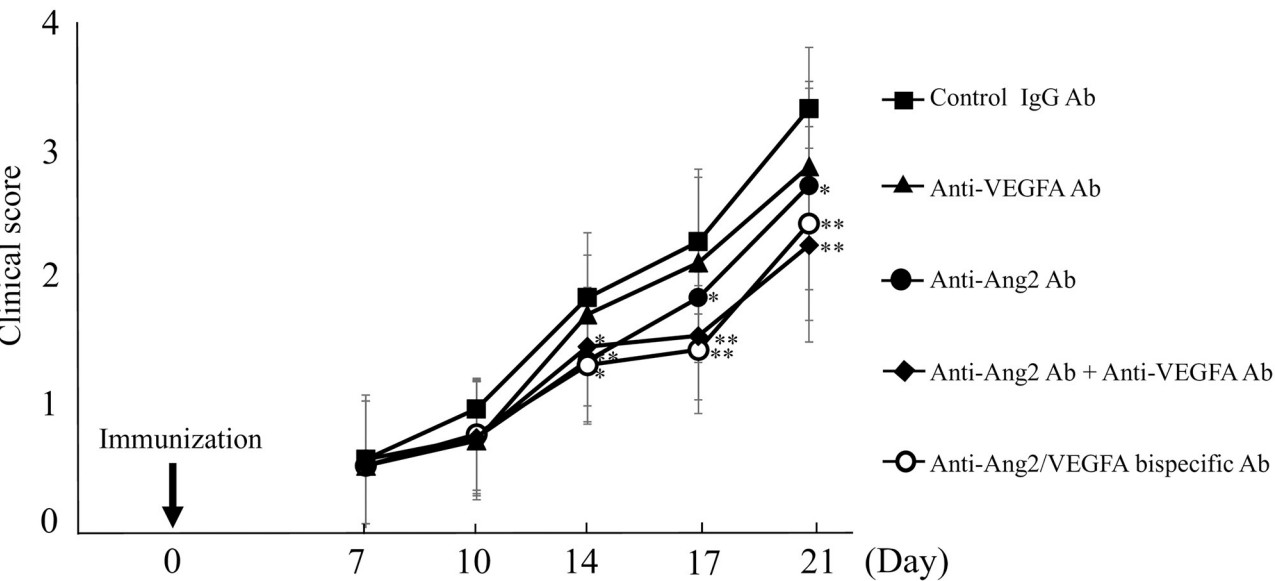

**Fig 4. Clinical scores of EAU mice.** The score of those treated with anti-Ang2 antibody was lower than that of those treated with control IgG antibody (n = 18–20 eyes per group, P<0.05). Furthermore, these scores were further decreased when treated by inhibiting both Ang2 and VEGFA (i.e., anti-Ang2 antibody with anti-VEGFA antibody treatment group and anti-Ang2/VEGFA bispecific antibody treatment group) (P<0.01). Although the score of those treated with anti-VEGFA antibody tended to be low, there was not a significant difference. Results are presented as Mean ± SD, and significance was determined by Kruskal-Wallis tests and Dunn procedure, *P<0.05, **P<0.01; compared with the control.

Ang2/VEGFA bispecific Ab (Fig 5B) mild vasculitis, with no retinal folds and no retinal detachments was observed. On the other hand, in EAU mice treated with control IgG Ab (Fig 5C) histopathology showed cell infiltration, vasculitis and granulomas in choroid and retina, demonstrating that treatment with anti-Ang2/VEGFA reduced the level of inflammation in the eye.

## Discussion

To our knowledge, this is the first report to suggest the involvement of Ang2 and VEGFA in the pathogenesis of ocular inflammatory diseases.

In the present study, we demonstrated three new findings concerning the role of Ang2 and VEGFA in the pathogenesis of ocular inflammation: (1) the high protein levels of Ang2 and VEGFA in vitreous samples of uveitis patients relative to controls, (2) the high expression of both Ang2 and VEGFA during the time course of the EAU mouse model relative to controls, and (3) the protective effects of dual Ang-2/VEGF-A inhibition in a preclinical mouse EAU study.

We demonstrated that the levels of Ang1, Ang2, VEGFA, and inflammatory cytokines were elevated in the vitreous samples of patients with uveitis compared to controls. In addition, the protein levels of Ang2 were higher than those of Ang1 in the patients with uveitis, whereas levels of Ang2 were lower than Ang1 in the controls, indicating a potential imbalance in the ratio of Ang2:Ang1 in patients with ocular inflammation. Previous studies reported that Ang2 and VEGF protein concentrations were increased in vitreous samples of patients with retinal diseases, such as retinopathy of prematurity and diabetic retinopathy [10,22,23], and also in the aqueous humor of patients with age-related macular degeneration [24,25]. Although it has been reported that the protein levels of VEGFA in vitreous samples are elevated in the patients

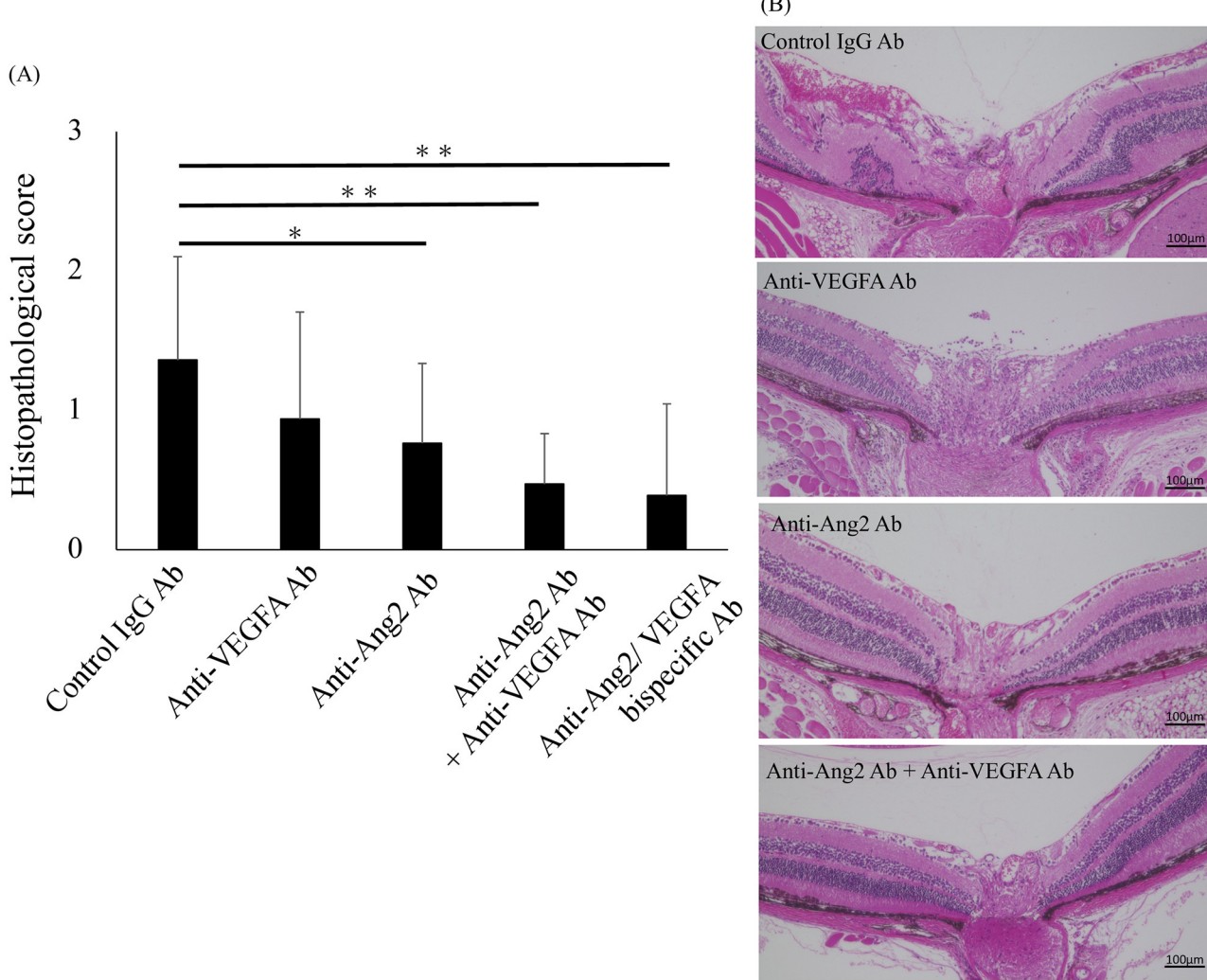

**Fig 5. Histopathological scores and histopathologies of EAU mice.** (A) The score of those treated with anti-Ang2 antibody was lower than that of those treated with control IgG antibody (n = 18–20 eye per group, P<0.05). Furthermore, these scores were further decreased when treated by inhibiting both Ang2 and VEGFA (i.e., anti-Ang2 antibody with anti-VEGFA antibody treatment group and anti-Ang2/VEGFA bispecific antibody treatment group) (P<0.01). Although the score of those treated with anti-VEGFA antibody had a trend towards being lower, there was not a significant difference. Results are presented as Mean ± SD, and significance was determined by Kruskal-Wallis tests and Dunn procedure, *P<0.05, **P<0.01; compared with the control. (B) Histopathology of EAU mouse treated with control IgG Ab results in cell infiltration, vascularitis and granulomas in choroid and retina, and these findings were reduced in EAU mouse treated with anti-VEGFA Ab or anti-Ang2 Ab. That of EAU mouse treated with anti-Ang2/VEGFA bispecific Ab results in mild vasculitis, no retinal folds and no retinal detachments.

with non-infectious uveitis [26], to our knowledge this is the first paper to reveal increased levels of Ang2 in vitreous samples of patients with uveitis. The elevated Ang2 and VEGFA in the eye suggest the involvement of both these molecules in the disease progression of uveitis patients.

In the serum samples, Ang1 levels in patients with uveitis were three times higher than in the controls. By contrast, Ang2 level showed no significant difference between the patients with uveitis and the controls. In a previous report, Choe *et al.* showed similar results in patients with Behçet's disease, including those with and without uveitis [27], although the cause of the difference between Ang1 and Ang2 remains unclear. When the serum level of Ang1 is high in

the patient with local inflammation, Ang1 may be induced in serum to inhibit Ang2 competitively. On the other hand, the increase of serum Ang2 has been shown in several systemic inflammatory diseases such as arthritis, sepsis, and cancer [28]. High Ang2 levels in serum may be seen when systemic inflammation occurs in the patients. The local inflammation may not be enough to increase inflammatory molecules in serum.

In this study, the protein levels of both Ang2 and VEGFA are high not systemically but locally in the ocular tissue. This result suggested that the main focus of inflammation was in ocular tissue and these molecules were involved in the pathogenesis of ocular inflammation.

To clarify the involvement of Ang2 and VEGFA in the pathogenesis of ocular inflammation, we examined the mRNA levels of Ang2 and VEGFA in the retinochoroidal tissues of EAU mice. The expression of VEGFA significantly increased at Day 11, when only slight EAU signs could be clinically observed, and showed a peak at Day 16. The expression of Ang2 also increased 16 days after immunization. In the retinochoroidal tissues of EAU mice, the high expression of VEGFA was followed by an increase in Ang2.

In the Ang-Tie2 signaling pathway, Ang2 is released from vascular endothelial cells by the stimulation of a range of stimuli, including VEGFA and TNF-α [29,30]. Our results are consistent with this theory, and we postulate that Ang2 and VEGFA are involved in the development of EAU. Although we have not examined the expression of TNF-α in this animal model, there are previous reports on the relation between TNF-α and uveitis [31]. Since Ang2 was elevated in human vitreous samples together with TNF-α in our study, it may be expected that TNF-α inhibition together with Ang2 and VEGFA may further suppress inflammation.

To determine whether Ang2 and VEGFA could be therapeutic targets for uveitis, we performed the treatments with each antibody in EAU mice.

Firstly, EAU was ameliorated by treatment with anti-Ang2 antibody monotherapy. Although previous reports suggested that the autoimmune inflammation was reduced using anti-Ang2 antibodies in several animal models of inflammation [11,12,14], this is the first study to reveal the efficacy of anti-Ang2 antibody for EAU. Ang1-Tie2 signaling downregulates inflammation by blocking NF-κB, and Ang2 exacerbates inflammation by competitively inhibiting Ang1-Tie2 signaling [8]. Ang2 also increases the expression of leukocyte adhesion molecules on endothelial cells via NF-κB by the indirect attenuation of anti-inflammatory Ang1-Tie2 signaling [32]. NF-κB is a well-known key regulator of inflammation and is a transcription factor controlling inflammatory genes that encode intercellular adhesion molecule-1, vascular cell adhesion molecule-1, and various proinflammatory cytokines. These adhesion molecules play key roles in the initial recruitment of leukocytes to the sites of inflammation. Therefore, NF-κB has also been discussed as a potential therapeutic target for inflammatory diseases, and future work as a follow up to these experiments could focus on understanding the contribution of NF-κB to the results reported here [33].

Other than this angiopoietin/Tie2 signaling pathway, Ang2 has Tie2-independent signaling pathway via β1 integrin which may be related to inflammation [34]. Previous studies reported that Ang2 induced endothelial cell migration and sprouting (i.e., angiogenesis) through focal adhesion kinase phosphorylation and Rac1 activation, and destabilize the endothelium through β1 integrin [35,36]. These pathways may also contribute to the EAU inflammation via Ang2, although further studies are required to elucidate the signaling pathways of Ang2 related to ocular inflammation.

Secondly, although the difference was not significant, the scores of EAU mice showed a trend toward being reduced by treatment with anti-VEGF antibody monotherapy. Previously, Anti-VEGF antibody treatment has been shown to reduce the number of inflammatory cells in a mouse model of psoriasis [37]. VEGFA is a ligand for VEGF receptor (VEGFR) 1, and the activation of VEGFR1 upregulates monocytes and macrophages migration via

phosphoinositide 3-kinase/Akt/NF-κB pathway or β2 integrin pathway [38–40]. In addition, the VEFGR1 activation in monocytes and macrophages upregulates the production of their pro-inflammatory cytokines included TNF-α and IL-6 [41,42].

Furthermore, the EAU scores of the dual-inhibition groups (i.e., the group treated with the combination of anti-Ang2 antibody and anti-VEGFA antibody and the group treated with Anti-Ang2/VEGFA bispecific antibody) showed a trend toward being lower than those of the single-inhibition groups, and greater significance levels versus control than anti-Ang2 mono-therapy. Previous study showed that the dual inhibition of Tie2 and VEGFA reduced disease severity and decreased macrophage infiltration in an animal model of collagen-indued arthritis [43]. The inhibition of Tie2 causes the blockade of not only Ang2 signaling but Ang1 signaling which can suppress inflammation. Ang2 inhibition may therefore be more ideal than Tie2 inhibition to regulate inflammation, however further experiments may be needed to explore this.

Ang2 and VEGFA are well-known molecules that play critical and coordinated roles in pathological angiogenesis and vascular leakage [9]. Faricimab (VABYSMO; Genentech/F. Hoffmann-La Roche Ltd), a bispecific anti-Ang2/VEGFA antibody that has two targets (one ligand-binding arm that binds Ang2 and the other that binds VEGFA), has recently been developed for treating patients with ocular neovascular diseases [10]. It was designed for intra-vitreal use and has shown improved anatomic outcomes and extended durability in patients with age-related macular degermation and diabetic macular edema respectively, and now is approved for use in multiple countries, as well as being studied in Phase 3 trials for retinal vein occlusion [44,45].

To optimize for ophthalmological use, its Fc region was engineered to abolish binding interactions with all Fc gamma recptor (FcγR) and the neonatal Fc receptor (FcRn). Elimina-tion of the binding of FcγR is highly desirable for the avoidance of unwanted inflammatory responses to therapeutic antibodies and fusion proteins. Therefore, the amino acids required for the FcγR interactions were exchanged to eradicate effector functions including antibody-dependent cellular cytotoxicity (ADCC), antibody-dependent cellular phagocytosis (ADCP), and complement-dependent cytotoxicity (CDC) in this bispecific antibody. Furthermore, by eliminating the FcRn binding site, IgG cannot be recycled and its systemic half-life should be reduced. This property can lead to high ocular, but low systemic exposure.

Although the antibodies used in our study did not have such modification of the Fc region, they showed sufficient anti-inflammatory effects. Our results suggest that, faricimab may be a promising therapeutical agent for ocular inflammation. However, a limitation of this EAU study, was that we started each treatment before the onset of EAU mouse model, not after the onset of inflammation, as would be the case in the human clinical setting. Therefore, future work would be needed to explore the therapeutic effects of combined anti-VEGF and anti-Ang2 as an intervention rather than prevention, at different time points.

In conclusion, we propose the simultaneous inhibition of Ang2 and VEGFA as a potential new therapeutic strategy for endogenous uveitis.

## Supporting information

**S1 Table. Protein levels of cytokines in the vitreous sample.**
(DOCX)

**S2 Table. Protein levels of cytokines in the serum sample.**
(DOCX)

**S3 Table. mRNA levels of cytokines in the EAU mice.**
(DOCX)

**S4 Table. Clinical scores of EAU mice.**
(DOCX)

**S5 Table. Histopathological scores of EAU mice.**
(DOCX)

## Acknowledgments

We thank Ikuyo Hirose and Shiho Yoshida (Department of Ophthalmology, Hokkaido University) for their technical assistance. We also wish to thank Stefan Dengl, Joerg Moelleken and their teams for the production and supply of antibodies for treatments in the animal experiments.

## Author Contributions

**Conceptualization:** Daiju Iwata, Kenichi Namba.

**Data curation:** Kayo Suzuki.

**Formal analysis:** Kayo Suzuki, Keitaro Hase, Miyuki Murata.

**Funding acquisition:** Daiju Iwata.

**Investigation:** Kenichi Namba, Keitaro Hase, Miyuki Murata.

**Methodology:** Kayo Suzuki, Kenichi Namba, Nobuyoshi Kitaichi.

**Project administration:** Daiju Iwata, Kenichi Namba, Miyuki Murata.

**Resources:** Daiju Iwata, Richard Foxton.

**Software:** Kenichi Namba, Miyuki Murata.

**Supervision:** Daiju Iwata, Kenichi Namba, Nobuyoshi Kitaichi, Susumu Ishida.

**Validation:** Miki Hiraoka, Nobuyoshi Kitaichi.

**Visualization:** Kayo Suzuki, Miki Hiraoka.

**Writing – original draft:** Kayo Suzuki.

**Writing – review & editing:** Daiju Iwata, Nobuyoshi Kitaichi, Richard Foxton, Susumu Ishida.

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
