## [Decision Letter · Decision Letter 0]

21 Jul 2023

PONE-D-23-12224Involvement of Angiopoietin 2 and vascular endothelial growth factor in uveitisPLOS ONE

Dear Dr. Iwata,

Thank you for submitting your manuscript to PLOS ONE. After careful consideration, we feel that it has merit but does not fully meet PLOS ONE’s publication criteria as it currently stands. Therefore, we invite you to submit a revised version of the manuscript that addresses the points raised during the review process.

We look forward to receiving your revised manuscript.

Kind regards,

Koichi Nishitsuka

Academic Editor

PLOS ONE

Journal Requirements:

"Initials of the authors who received each award is D.I. Grant numbers awarded to each author are 3. The names of funders are  JSPS KAKENHI [grant number 16K29294] and J&J surgical vision (AMO) [grant number 

AS2021A000044497]. URLs are as follow. https://www.jsps.go.jp/j-grantsinaid/

https://surgical.jnjvision.jp/clinical%26medical/research-grant

any role; No."

5. Please expand the acronym “JSPS, J&J” (as indicated in your financial disclosure) so that it states the name of your funders in full.

6. Thank you for stating the following in the Competing Interests section: 

"The authors declare no competing interests except for RF who is an employee of F. Hoffmann-La Roche Ltd."

We note that one or more of the authors are employed by a commercial company: Hoffmann-La Roche Ltd.

(2) Please also provide an updated Competing Interests Statement declaring this commercial affiliation along with any other relevant declarations relating to employment, consultancy, patents, products in development, or marketed products, etc.  

Within your Competing Interests Statement, please confirm that this commercial affiliation does not alter your adherence to all PLOS ONE policies on sharing data and materials by including the following statement: ""This does not alter our adherence to  PLOS ONE policies on sharing data and materials.” (as detailed online in our guide for authors http://journals.plos.org/plosone/s/competing-interests).

If this adherence statement is not accurate and  there are restrictions on sharing of data and/or materials, please state these. Please note that we cannot proceed with consideration of your article until this information has been declared.

7.  In your Data Availability statement, you have not specified where the minimal data set underlying the results described in your manuscript can be found. PLOS defines a study's minimal data set as the underlying data used to reach the conclusions drawn in the manuscript and any additional data required to replicate the reported study findings in their entirety. All PLOS journals require that the minimal data set be made fully available. For more information about our data policy, please see http://journals.plos.org/plosone/s/data-availability.

8. We note that you have stated that you will provide repository information for your data at acceptance. Should your manuscript be accepted for publication, we will hold it until you provide the relevant accession numbers or DOIs necessary to access your data. If you wish to make changes to your Data Availability statement, please describe these changes in your cover letter and we will update your Data Availability statement to reflect the information you provide.

Reviewers' comments:

Reviewer's Responses to Questions

**Comments to the Author**

1. Is the manuscript technically sound, and do the data support the conclusions?

Reviewer #1: Partly

Reviewer #2: Yes

2. Has the statistical analysis been performed appropriately and rigorously? 

Reviewer #1: No

Reviewer #2: Yes

3. Have the authors made all data underlying the findings in their manuscript fully available?

Reviewer #1: Yes

Reviewer #2: Yes

4. Is the manuscript presented in an intelligible fashion and written in standard English?

Reviewer #1: Yes

Reviewer #2: Yes

5. Review Comments to the Author

Reviewer #1: The authors detected the expression level of some cytokines, such as Ang-1, Ang-2, VEGFA in the vitreous samples and serum. Also, they investigated the effect of anti-ang2 antibody and anti-VEGFA antibody on the EAU and they suggested that ang-2 and VEGFA might be involved in the uveitis. However, the authors should provide sufficient evidences to support the conclusion.First, the expression level of Ang-2 and VEGFA of the vitreous and serum is not consistent with serum. The expression level of ang-2 and VEGFA in vitreous samples are higher however is no different in serum. So why did the authors choose the Ang-2 and VEGFA as targets for treatment? In the treatment section of EAU, the pictures of clinical findings and the histopathological figures of the anti-ang2-treated eye balls and anti-VEGFA-treated eye balls should be provided. In addition, the scientific meaning in the study was unclearly described in the abstract.

Reviewer #2: The study has novel findings about a possible role of Ang2 and VEGFA in the process of uveitis. The findings may lead to

new opinions on treatment as they clearly have stated. They have found an increased level of expression/concentration of these agents. These may revolutionize pathophysiological concepts.

I have one concern: to obtain vitreous samples from 16 patients with idiopathic uveitis, the authors have performed pars plana vitrectomy -an aggressive procedure with potential complications- in cases that might not been candidated for it routinely. It is a serious ethical concern. Please explain.

6. PLOS authors have the option to publish the peer review history of their article (what does this mean?). If published, this will include your full peer review and any attached files.

Reviewer #1: **Yes: **Feilan Chen

Reviewer #2: No

---

## [Author Response · Author response to Decision Letter 0]

5 Oct 2023

We appreciate your careful review of our manuscript and the guidance provided for revision. As requested, we have revised the manuscript, which contains significant corrections according to the guidance provided. The changes made have been highlighted using the "track changes" function of Microsoft Word. Please find below our point-by-point responses to the reviewers’ comments reproduced in italics.

---

## [Decision Letter · Decision Letter 1]

8 Nov 2023

Involvement of Angiopoietin 2 and vascular endothelial growth factor in uveitis

PONE-D-23-12224R1

Dear Dr. Iwata,

We’re pleased to inform you that your manuscript has been judged scientifically suitable for publication and will be formally accepted for publication once it meets all outstanding technical requirements.

Kind regards,

Koichi Nishitsuka

Academic Editor

PLOS ONE

Additional Editor Comments (optional):

Reviewers' comments:

Reviewer's Responses to Questions

**Comments to the Author**

1. If the authors have adequately addressed your comments raised in a previous round of review and you feel that this manuscript is now acceptable for publication, you may indicate that here to bypass the “Comments to the Author” section, enter your conflict of interest statement in the “Confidential to Editor” section, and submit your "Accept" recommendation.

Reviewer #3: All comments have been addressed

2. Is the manuscript technically sound, and do the data support the conclusions?

Reviewer #3: Yes

3. Has the statistical analysis been performed appropriately and rigorously? 

Reviewer #3: Yes

4. Have the authors made all data underlying the findings in their manuscript fully available?

Reviewer #3: Yes

5. Is the manuscript presented in an intelligible fashion and written in standard English?

Reviewer #3: Yes

6. Review Comments to the Author

Reviewer #3: I have appreciate to authors for their satisfied corrections and prompt responses. I'm happy to accept this manuscript.

7. PLOS authors have the option to publish the peer review history of their article (what does this mean?). If published, this will include your full peer review and any attached files.

Reviewer #3: **Yes: **Madhu Sudhana Saddala

---

## [Editor Report · Acceptance letter]

16 Nov 2023

PONE-D-23-12224R1 

Involvement of Angiopoietin 2 and vascular endothelial growth factor in uveitis 

Dear Dr. Iwata:

I'm pleased to inform you that your manuscript has been deemed suitable for publication in PLOS ONE. Congratulations! Your manuscript is now with our production department. 

Kind regards, 

on behalf of

Dr. Koichi Nishitsuka 

Academic Editor

PLOS ONE